# The Significance of Fungal Specialized Metabolites in One Health Perspectives

**DOI:** 10.3390/ijms26073120

**Published:** 2025-03-28

**Authors:** Pierluigi Reveglia, Carmela Paolillo, Gaetano Corso

**Affiliations:** Department of Clinical and Experimental Medicine, University of Foggia, 71122 Foggia, Italy; pierluigi.reveglia@unifg.it (P.R.); carmela.paolillo@unifg.it (C.P.)

**Keywords:** fungal metabolites, One Health, metabolomics

## Abstract

Among the emerging threats in global health, fungal pathogens stand out as some of the most important, causing over 1.6 million deaths annually and destroying a third of all food crops each year, exacerbating food insecurity and economic losses. Climate change further amplifies the threat by enabling pathogenic fungi to survive at mammalian temperatures, increasing risks of zoonotic transmission and antifungal resistance. In this context, interdisciplinary research, particularly the One Health approach, is crucial for understanding the evolution of fungal resistance and improving diagnostic and therapeutic tools. Drawing lessons from agriculture, where integrated pest management strategies successfully mitigate fungal threats, could offer new ways to tackle fungal infections in humans. Advanced metabolomics and diagnostics, including fungal metabolites as biomarkers, hold promise for early detection and personalized treatment. Collaborative efforts between medicine, veterinary science, and plant pathology are essential to develop new antifungal drugs and improve clinical management of fungal diseases, fostering a more resilient global health system.

## 1. Introduction

In the last few decades, a significant enhancement in the appearance of infectious agents has been observed, among which SARS-CoV-2 is only the last example that further highlighted the limits of reductionism in studying complex phenomena where interactions are relevant [1]. Indeed, human-driven ecosystem disruption has created the ideal conditions for zoonotic spillovers. Pathogenic fungi are responsible for over 1.6 million deaths annually, with more than one billion people suffering from the most common fungal infections [2]. Moreover, the impact of fungal pathogens on human health goes beyond the ability of fungi to infect humans, as they destroy a third of all food crops annually, causing economic losses and negatively impacting agriculture [2]. The rise in global temperature due to global warming could facilitate the survival of pathogenic fungi at mammalian temperatures. Fungi, usually involved in plant diseases, which have also begun to be reported as human pathogens, strengthen this threat. In late 2022, the World Health Organization (WHO) published the fungal priority pathogens list, highlighting fungi of critical or high importance to human health and outlining pathways for action [3]. Unfortunately, the antifungal drug development and approval process is often challenging and slow, necessitating an urgent collaborative approach that spans human medicine, veterinary medicine, and plant pathology. This approach should emphasize antifungal stewardship, raise awareness among clinicians and the general public, involve industry, and increase laboratory capacity to detect and monitor antifungal drug resistance in humans, animals, and the environment [4,5].

In recent years, the concept of One Health has taken hold, integrating knowledge from medicine, biology, and chemical ecology. The One Health High-Level Expert Panel (OHHLEP) recently amended the definition as follows: “One Health is an integrated, unifying approach that aims to sustainably balance and optimize the health of people, animals and ecosystems. It recognizes the health of humans, domestic and wild animals, plants, and the wider environment (including ecosystems) are closely linked and inter-dependent. The approach mobilizes multiple sectors, disciplines and communities at varying levels of society to work together to foster well-being and tackle threats to health and ecosystems” [6]. This approach is particularly important for understanding the evolution of antimicrobial resistance and improving diagnostic tools [5]. The study of microbial ecology can be crucial in promoting sustainable development in One Health, in particular.

## 2. Learning from Agriculture

One of the critical challenges in both agriculture and medicine is the threat posed by mycotoxins, specialized metabolites produced by certain fungi that contaminate food and pose serious health risks. Among the most concerning mycotoxins are aflatoxins, ochratoxins, and fumonisins, structural examples of which are shown in Figure 1, all of which have significant implications for human health [7]. Aflatoxins, primarily produced by Aspergillus species, are highly carcinogenic and have been linked to liver cancer, immune suppression, and developmental issues. Ochratoxins, produced by *Aspergillus* and *Penicillium* species, are nephrotoxic, impacting kidney function and potentially contributing to chronic kidney disease. Fumonisins, produced by Fusarium species, are associated with neural tube defects and esophageal cancer. Farmers and agricultural scientists have developed protocols for mycotoxin management, which include rigorous monitoring and environmental controls [8]. Thus, medicine can learn valuable lessons from agriculture, where managing fungal pathogens provides a strong framework for addressing similar threats to human health.

Consider the concept of integrated pest management (IPM), a pillar of the agricultural struggle against pests, including fungi. This comprehensive strategy combines biological, cultural, physical, and chemical methods to mitigate crop economic and environmental risks [9]. Adopting a similar multidimensional approach—integrating pharmacological interventions with lifestyle and environmental adjustments—could establish a more effective defense against fungal infections, especially in the context of rising antifungal resistance. Plant breeding is another strategy to gain resilience against biotic stress [10]. Similarly, comprehending the genetic factors conferring resistance to fungal infections in certain humans may pave the way for personalized medical interventions. Furthermore, farmers have learned how to adjust the crop environment to avoid the proliferation of fungi. This care for the environment is also vital in medicine, where managing hospital environments—air quality, humidity, and other factors—could substantially reduce the incidence of hospital-acquired fungal infections. The role of education is also fundamental, as scientists and healthcare professionals can equip communities with the knowledge and tools needed to prevent, recognize, and effectively treat fungal diseases, much like plant pathologists help farmers manage crop diseases.

## 3. Diagnostics of Fungal Metabolites

Agriculture’s vigilance in the early detection and surveillance of fungal pathogens, employing cutting-edge technologies to limit potential outbreaks, offers a lesson in proactive responsiveness. However, frequently, human infection remains undiagnosed. For instance, since early 2021, cases of COVID-19-associated mucormycosis have increased, particularly among patients with uncontrolled diabetes. Diagnosis of COVID-19-associated mucormycosis is challenging, as the clinical and radiological features of pulmonary and disseminated mucormycosis are non-specific, resulting in missed or late diagnoses. The most common form of this infection is rhino-orbital cerebral mucormycosis, which carries a mortality rate of 49%, especially in cases involving the lungs or brain. Those who survive often face severe complications, such as vision loss in 46% of cases. The disease is frequently undiagnosed in India, contributing to its high rates of morbidity and mortality, particularly in patients with pulmonary involvement [11]. Enhancing diagnostic tools and monitoring systems for early detection could have a profound impact on medical outcomes.

Fungi have independently evolved the ability to infect humans multiple times over millions of years. A recent review provided a comprehensive overview of the pathobiology of human fungal infections [12]. Nearly half of all fungal phyla contain species that can cause disease, with new fungal pathogens emerging due to increased human exposure, weakened immune systems, and fungi adapting to new environments. Morphological plasticity is essential for virulence, as many fungi can change forms to enhance their ability to infect. A key aspect of fungal pathogenicity is the secretion of molecules that aid in nutrient acquisition, immune evasion, and the infection process. Pathogenic fungi typically possess more genes related to secreted factors compared to their non-pathogenic relatives. Furthermore, fungi produce small-molecule secondary metabolites, such as penicillin, which serve various functions, including immune modulation and micronutrient scavenging. They also release extracellular vesicles (EVs), which may play a role in intercellular communication and the progression of disease. Despite their significant impact on human health, fungi are still not well understood, and many of their mechanisms of infection and interactions with human physiology are still under investigation.

Recent studies emphasize the vital role of antibody immunity in combating fungal infections and the importance of understanding human–fungal pathogen interactions. Antigenic proteins stimulate antibody production and are crucial for developing therapeutic strategies. Immunoproteomics has enabled the identification of these proteins in various fungi, revealing their roles in stress responses and metabolism. These proteins offer opportunities for creating species-specific or broad-spectrum diagnostic tools, therapeutic antibodies, and vaccines. While research on antigenic proteins enhances our understanding of host–pathogen interactions, the potential to target fungal specialized metabolites and mycotoxins with antibodies remains largely unexplored, representing a significant opportunity for mitigating their harmful effects [13].

Another fascinating investigation field related to human–fungal pathogen interaction is the immunity of the central nervous system (CNS) to fungal infections, a poorly understood area of medical mycology, despite brain-tropic fungi causing many fatalities. Recent research has improved our understanding of neuroinflammation and the immune system’s role in CNS tissue homeostasis, revealing mechanisms of fungal invasion, microglia–astrocyte interaction, and the regulation of adaptive immune responses [14]. Fungi are also linked to non-infectious CNS diseases like multiple sclerosis, Parkinson’s disease, Alzheimer’s disease, and amyotrophic lateral sclerosis. These organisms elicit unique inflammatory responses that may worsen disease progression, suggesting they could become therapeutic targets [15]. The frequency of CNS mycoses has risen dramatically due to there being more immunocompromised individuals, yet treatment options are limited. Research shows that CNS-resident cells, including microglia and astrocytes, detect fungal pathogens and initiate immune responses, challenging the view of the CNS as immune-privileged. Understanding these host–pathogen interactions is crucial for developing new therapies for CNS fungal infections [16].

Various modern, high-throughput, and rapid techniques are used for surveillance and diagnosis, including DNA-based methods like MinION, SmidgION, and Loop-Mediated Isothermal Amplification (LAMP). Additionally, biochemistry and analytical chemistry techniques are employed to identify biomarkers, including proteins, lipids, and specialized metabolites, produced by microorganisms [17]. Studies have shown that metabolite profiles can be used in combination with DNA-based methods to improve the sensitivity and specificity of fungal pathogen detection, which is crucial in immunocompromised patients or when diagnosing invasive infections. Leveraging metabolomics alongside genomics, as seen in microbiome research, offers enhanced insight into pathogen–host interactions, which can directly lead to more effective diagnostic tools [18,19]. Fungal specialized metabolites offer promising potential as diagnostic biomarkers for a range of diseases, particularly rare fungal infections [20]. For example, metabolites produced by fungi such as *Candida* and *Aspergillus* species can be detected in blood, tissue, and respiratory samples, allowing for early diagnosis before more invasive clinical symptoms arise. The incidence of *Aspergillus*-related infections has surged, largely due to increased use of immunosuppressive therapies, and these infections are associated with a high mortality rate. Unfortunately, laboratory diagnosis of *Aspergillus* infections remains difficult. A recent study compared the metabolomic profiles of 30 different strains that originated from 6 Aspergillus species and 10 non-*Aspergillus* fungi. The authors were able to identify eight *Aspergillus*-specific compounds that could be potential diagnostic biomarkers for pathogenic infection [21]. Moreover, in a prospective study, researchers found that bis(methylthio)gliotoxin (bmGT), a secondary metabolite of *Aspergillus*, could be a promising biomarker for invasive aspergillosis (IA). When compared to the widely used galactomannan (GM) test in 79 patients, bmGT demonstrated higher sensitivity and a positive predictive value (PPV) while maintaining similar specificity. Combining bmGT with GM improved diagnostic accuracy, achieving a PPV of 100% and a negative predictive value (NPV) of 97.5%, suggesting its potential in guiding antifungal treatment [22]. In another study, gas chromatography and mass spectrometry were employed to identify volatile organic compounds (VOCs) that can distinguish between *Candida albicans*, *C. glabrata*, *C. krusei*, and *C. tropicalis*. By analyzing the specific VOC profiles of each species, the researchers were able to develop a non-invasive method to differentiate these fungal pathogens, which could be applied to enhance early diagnosis and species-specific identification in clinical settings [23]. This approach holds promise for early diagnosis and monitoring of fungal infections. Identifying unique metabolite profiles associated with specific fungal pathogens could facilitate the development of rapid, non-invasive diagnostic tools [24].

These biomarkers can be integrated into existing platforms, such as detection of free nucleic acids in biological fluids, to improve diagnostic sensitivity and specificity. Such advancements in fungal metabolite-based diagnostics could dramatically improve clinical outcomes by enabling timely intervention and reducing the severity of fungal infections, especially in immunocompromised patients [25]. Moreover, few to no studies comparing the ex vivo and in vivo sampling of pathogens and hosts during infection have been conducted. An example of this type of study focused on the in vitro analysis of the metabolomic profile during infection of lung epithelial cells by *C. neoformans* [26]. The authors simulated the lung infection using a culture supernatant of lung epithelial cells infected with *C*. *neoformans* at low and high multiplicities of infections (MOIs). The results showed that 10 discriminative metabolites were identified during *C. neoformans* infection in lung epithelial cells at different infection loads. L-cysteine and pantothenic acid were found in both loads, indicating disruptions in the glyoxylate and dicarboxylate cycles. Pantothenic acid enhanced *C. neoformans* growth and biofilm viability, while adonitol increased biofilm cell viability. Disruptions in β-alanine metabolism at higher infection loads suggest that further investigation is needed, particularly in animal models, to understand its role during infection [27].

## 4. Therapeutic Applications of Fungal Metabolites

The isolation and characterization of bioactive natural products, and in particular specialized metabolites, have a significant impact in many fields, such as agriculture, chemical ecology, and medicine, encouraging the creation of professional multidisciplinary networks between scientists, paving the way for a more systemic investigation approach, as required by the One Health concept. Indeed, recent advances in molecular science have enhanced our ability to engineer these pathways, allowing for the design and synthesis of specialized metabolites in a single step. Techniques such as gene deletion, addition, and tailored enzyme engineering enable the creation of new compounds. These advancements can efficiently convert low-value materials into high-value products when combined with fungi that can metabolize waste. Advanced molecular tools have made rapid cloning and synthetic DNA assembly standard practices. Affordable genome sequencing and user-friendly bioinformatics have streamlined fungal biotechnology, expanding the potential of these organisms [27]. In this framework, the role played by chemists, especially the chemists involved in the investigation of the isolation and characterization of natural compounds by applying metabolomics, is fundamental to help revive a discipline with engaging, novel challenges [28]. Indeed, untargeted metabolomics offers an opportunity to explore the chemical space with a high throughput and without bias, merging the gap between chemistry and systems biology. Nuclear Magnetic Resonance (NMR) and high-resolution chromatography coupled with tandem mass spectrometry (LC-MS/MS) are the two techniques mainly used in metabolomics. Due to the complexity of the data, chemometric analysis is necessary to highlight biological variability in the samples. Additionally, other environmental factors influence the production of specialized metabolites, such as nutrient availability, temperature, and pH, and the One Strain–Many Compounds (OSMAC) approach has emerged as a powerful tool for exploring the chemical diversity of fungal metabolites [29]. Marine fungi are a vital source of lead compounds, offering promising opportunities for drug discovery and targeted therapies due to their unique metabolic pathways and biodiversity. Notably, species of *Penicillium* and *Aspergillus* produce bioactive compounds found in sponges, coral, and seawater. Several reviews have examined research trends related to natural products derived from marine fungi, emphasizing that these compounds primarily belong to the polyketide class, which exhibit potent antiviral, antitumor, and antibacterial properties [30,31].

In recent research, several new fungal metabolites with promising clinical applications have been identified. These metabolites have potential drug targets, offering new avenues for the development of antifungal therapies and treatments for other diseases. The discovery of fungal metabolites with potent bioactivity opens the door to novel therapeutic avenues, particularly in developing new antifungal drugs. With rising antimicrobial resistance, there is an urgent need for new classes of drugs that target fungal pathogens more effectively. Some fungal metabolites have already shown promising results as drug candidates due to their unique mechanisms of action, which differ from those of traditional antifungal agents [32]. For example, fungal-derived compounds like echinocandins have revolutionized antifungal therapy by targeting the fungal cell wall [33]. Expanding the search for bioactive metabolites in fungi could lead to the discovery of additional therapeutic compounds that can be developed into next-generation antifungal agents. Moreover, such metabolites could serve as scaffolds for synthetic modification, enhancing their efficacy and minimizing adverse effects. Their unique biochemical properties could be harnessed to design drugs that are more effective and have fewer side effects than current treatments. The potential utility of these metabolites in predictive medicine is fascinating. By understanding the metabolic profiles associated with specific diseases, we can develop predictive models that anticipate disease progression and response to treatment. This approach can lead to more personalized medical care, tailoring interventions to the individual’s metabolic makeup and improving the efficacy of treatments.

Moreover, fungi, and in particular endophytic fungi, are the source of several drugs used to treat many chronic conditions other than chronic infection (antifungal, antiviral, and antibiotic drugs). Indeed, endophytic fungi have gained significant attention in recent years for their potential to revolutionize the development of therapeutic agents. These ubiquitous microorganisms reside symbiotically within plants’ epidermal and aerial tissues, functioning as silent chemists in the natural world. Endophytic fungi have become invaluable allies in combating a broad spectrum of human health threats by producing a variety of biologically active secondary metabolites, including terpenes, alkaloids, monoterpenoids, peptides, and polyketides. The versatility of these fungi stems from their unique metabolic pathways, which facilitate the isolation and application of diverse bioactive compounds in medicine. Researchers advocate not only the expansion of endophyte applications but also refining formulation methods to ensure that these solutions are safer, more accessible, cost-effective, and efficient. The untapped microbial diversity within endophytic fungi represents a treasure trove of novel bioactive compounds that could drive biotechnological advancements. By accelerating the screening and discovery of new biomolecules, scientists aim to tackle life-threatening diseases, safeguarding human health and advancing the medical and pharmaceutical sectors [34].

Major classes of fungal metabolites, such as anthraquinones, sesquiterpenoids, chromones, xanthones, phenols, and cyclic peptides, have demonstrated profound biological activity. Advances in genomic analysis reveal that many fungi possess latent gene clusters capable of producing yet-undiscovered secondary metabolites. These “cryptic” biosynthetic gene clusters can be activated under specific conditions, such as stress or exposure to small-molecule epigenetic modifiers. Through chemical–epigenetic regulation—using DNA methyltransferase and histone deacetylase inhibitors—scientists have successfully induced or enhanced the production of over 540 secondary metabolites, many of which exhibit antimicrobial, anticancer, anti-inflammatory, and antioxidant properties [35].

Fungal metabolites are currently used or in clinical trials as anticancer agents, immunosuppressants and immunomodulators, central nervous system-disease-related agents, and statins. While the consumption of fungi as medicine and hallucinogens has been a part of many cultures for centuries, the modern era of fungal drug discovery is often traced back to the discovery of penicillin in 1928. Alexander Fleming isolated penicillin C from *Penicillium rubens*, a beta-lactam antibiotic capable of inhibiting the cross-linking of peptidoglycans in the bacterial cell wall. The next major fungi-derived antibiotic was cephalosporin C, isolated from *Acremonium chrysogenum* in 1945, which shares the same mechanism of action as penicillin.

Since then, other bioactive metabolites have been developed and commercialized, and several reviews have been written on this topic [36,37]. Notable examples are described below, and their structures are reported in Figure 2 and Figure 3. Penicillenols (**1**), derived from *Penicillium* sp., exhibit potent cytotoxicity against numerous cell lines, demonstrating their potential in anticancer therapies. Similarly, taxol (**2**), isolated from *Taxomyces andreanae*, stands as one of the most effective and groundbreaking anticancer drugs discovered from endophytic fungi, revolutionizing cancer treatment with its ability to stabilize microtubules and inhibit cell division. Several other compounds highlight the antimicrobial potential of fungal metabolites. Clavatol (**3**), produced by *Torreya mairei*, along with sordaricin (**4**) from *Fusarium* sp., jesterone (**5**) from *Pestalotiopsis jesteri*, and javanicin (**6**) from *Chloridium* sp., possess potent antibacterial and antifungal properties, making them effective against a range of foodborne infectious agents [37]. Their ability to combat microbial pathogens underscores their importance in food safety and medical applications. The antioxidant potential of fungal metabolites is exemplified by pestacin (**7**), isolated from *Pestalotiopsis microspora*, which exhibits outstanding free-radical scavenging properties. Such antioxidant activity prevents cellular damage and manages oxidative stress-related conditions. Beyond these, several bioactive compounds—such as camptothecin (**8**), diosgenin (**9**), hypericin (**10**), podophyllotoxin (**11**), vinblastine (**12**), and paclitaxel (**13**)—have been commercially developed from endophytic fungi, with applications spanning agriculture and pharmaceuticals. These compounds often mimic or enhance the activity of phytohormones and essential oils, emphasizing their multifaceted significance.

The genus *Aspergillus* presents an intriguing paradox. Although classified as a health threat by the World Health Organization [3], due to its pathogenic potential, it is also a prolific producer of bioactive metabolites [38]. For instance, antibacterials such as aspochalasin P (**14**), alatinone (**15**), 11-methoxycurvularine (**16**), and 12-keto-10,11-dehydrocurvularine (**17**) have been purified from *Aspergillus* sp., showcasing its dual nature. Additionally, aspergillone A (**18**), a newly discovered polyketide from *Aspergillus cristatus* associated with *Pinellia ternata*, represents a milestone in fungal metabolite research. Aspergillone A exhibits antibacterial activity against *Bacillus subtilis* and *Staphylococcus aureus*, with MIC50 values of 8.5 µg/mL and 32.2 µg/mL, respectively. Its structure—a unique bicyclo [2.2.2]diazaoctane indole alkaloid constructed from tryptophan and alanine—offers a novel chemical framework for further exploration [37,38].

Two particularly promising compounds, ophiobolin A (**19**) and sphaeropsidin A (**20**), have captured the scientific community’s attention. Initially identified as phytotoxins, these metabolites have demonstrated powerful anticancer activities. With its unique 5:8:5 ring structure, Ophiobolin A has shown efficacy in targeting malignant cancers such as glioblastomas and melanomas. Meanwhile, sphaeropsidin A exhibits versatile biological properties, including antibiotic, antifungal, antiviral, and anticancer activities, underscoring its potential as a multipurpose therapeutic agent. Ongoing research into their structural and functional modifications continues to yield derivatives with enhanced biological activity [39].

Finally, fungal compounds have also been a good source of immunosuppressant agents. One example is mycophenolic acid (**21**), isolated for the first time from *Penicillium brevicompactum*, which blocks lymphocyte proliferation and is used as an immunosuppressant for organ transplantation surgery [40]. Commercially available synthetic derivates of mycophenolic acid have been approved since 1995.

In essence, the diverse metabolites produced by fungi present a valuable opportunity for discovering innovative therapeutic agents. Their ability to create specialized compounds should encourage the scientific community to further invest in research on fungal biodiversity and its promising applications in medicine, agriculture, and other fields.

## 5. Personalized Medicine and Fungal Metabolites

The integration of metabolomics into personalized medicine is rapidly evolving, with fungal metabolites playing a critical role in tailoring treatments for patients. Translational research has shown that fungal metabolite profiles can predict disease progression and treatment response, providing a pathway to personalized therapeutic strategies. For example, specific metabolite signatures can indicate resistance to certain antifungal therapies, guiding clinicians in selecting the most effective treatments for individual patients. Moreover, the integration of metabolomics with genomics and proteomics (multi-omics approaches) allows for a comprehensive understanding of host–pathogen interactions, paving the way for precision medicine in the management of fungal infections [41,42,43]. The potential for personalized medicine extends beyond diagnostics and can influence therapeutic decision-making, ultimately improving patient outcomes and reducing the risk of treatment failure.

Nevertheless, the vast chemical diversity and complexity of fungal metabolites pose a significant challenge for untargeted metabolomics using LC-MS/MS in assigning structures to metabolites of interest without bias. Despite significant progress, many signals detected in metabolomics experiments cannot be directly assigned to specific metabolites due to the absence of their spectra in metabolomics databases, particularly for fungal metabolites. Modern dereplication strategies have been developed to address this issue, including machine learning, in silico fragmentation, and molecular networking [29]. Finally, incorporating taxonomical information or metadata from previous biological knowledge and employing a combination of chemoinformatic tools can aid in accurately dereplicating fungal metabolites.

Traditional metabolomics approaches have limitations when studying the interaction between different organisms. Alwood and colleagues [44] suggest a dual approach that considers metabolites produced by both the pathogen and the host. Additionally, considering the interaction between metabolites, such as synergy, as proposed by Vidar and colleagues [45], could lead to a better understanding of the pathosystem. A dual approach is necessary to recognize that a pathogenic attack involves an interaction between two partners that can only be partially assessed if focusing only on one partner. Regarding metabolomics, this reciprocal interaction will include metabolites produced by one partner being excreted and subsequently absorbed by the other (metabolite cross-talk or the interactome). Additionally, each partner releases metabolites into the extracellular environment, further influencing the metabolic dynamics of the interaction. Therefore, when studying host–pathogen interactions and searching for biomarkers, it is vital to focus on both sides and conduct an initial analysis of the microbe’s metabolome. Furthermore, focusing on specialized metabolites and combining them with classic DNA detection techniques, such as qPCR, is essential. Indeed, detecting phytotoxins could indicate that an infection is in progress and that a management procedure should be applied. Early-detection biomarkers could be pathogen metabolites that accumulate in the host tissues and can only be identified by knowing the specialized metabolite produced by the pathogen.

By leveraging metabolomics approaches to identify and validate such biomarkers, clinicians can enhance their ability to diagnose and manage rare fungal infections effectively, ultimately improving patient outcomes and reducing the burden of these challenging conditions. Continued research efforts in this direction hold great promise for advancing precision medicine approaches and addressing the unmet needs of individuals affected by rare fungal diseases.

## 6. Conclusions and Further Directions

In conclusion, studying specialized metabolites is crucial for understanding the pathogenicity and virulence of specific pathogens, as well as for developing more sensitive diagnostic tools and sustainable control methods. Given the increasing incidence of fungal infections worldwide, there is a pressing need for interdisciplinary research. Exchanging ideas and strategies could initiate a new era of innovation and resilience, fostering the One Health approach. Metabolomics must be further integrated with other omics disciplines. These studies should focus on deciphering the metabolic changes occurring in both the host and the pathogen during their interaction. Through this approach, novel biomarkers and potential drug candidates with pharmacological activity can be identified and further explored, potentially leading to new therapeutic options and better clinical management of fungal infections.

Lastly, legislators should prioritize interdisciplinary research to address the issues mentioned above. They should allocate additional funding to projects exploring this niche scientific field, which is expected to see significant growth in the coming years.

## Figures and Tables

**Figure 1 ijms-26-03120-f001:**
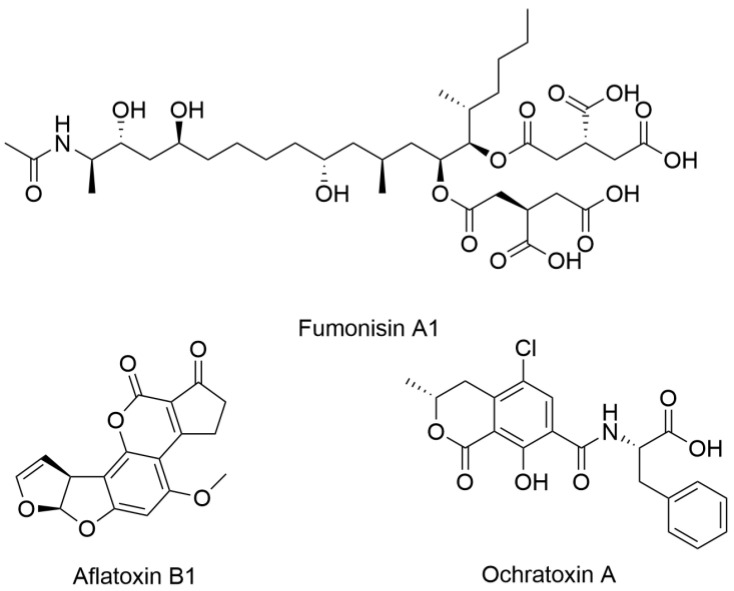
Structural examples of aflatoxins, ochratoxins, and fumonisins.

**Figure 2 ijms-26-03120-f002:**
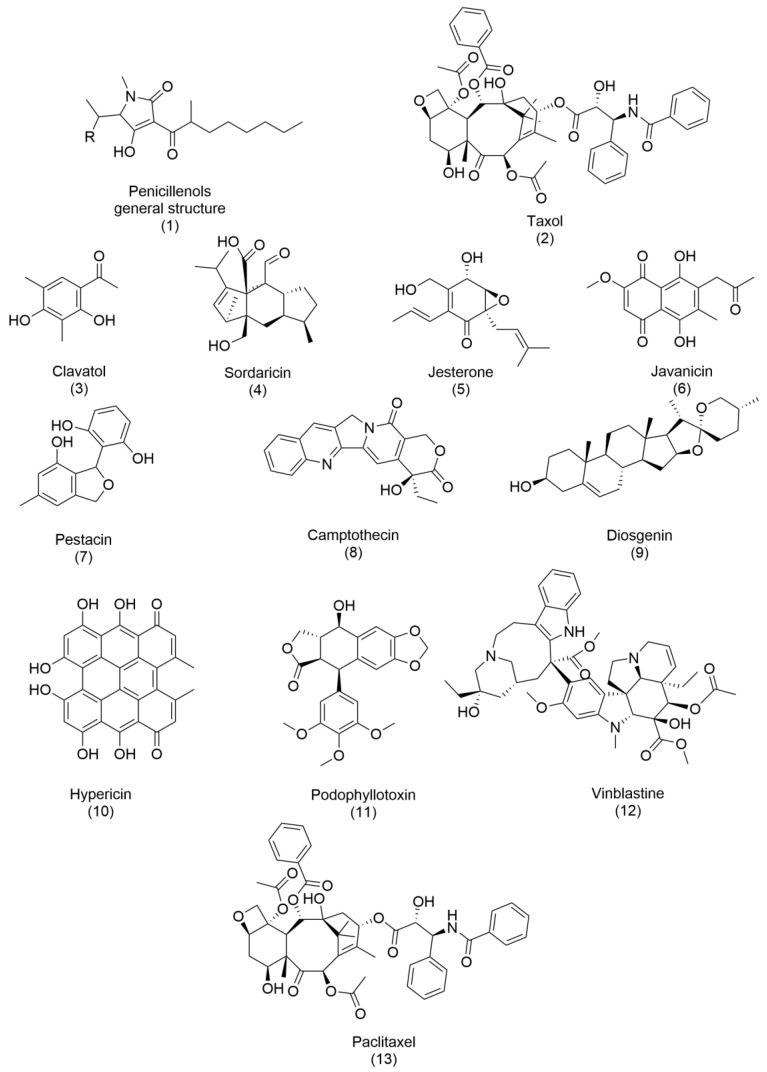
Structures of: penicillenols (**1**), taxol (**2**), clavatol (**3**), sordaricin (**4**), jesterone (**5**), javanicin (**6**), pestacin (**7**), camptothecin (**8**), diosgenin (**9**), hypericin (**10**), podophyllotoxin (**11**), vinblastine (**12**), and paclitaxel (**13**).

**Figure 3 ijms-26-03120-f003:**
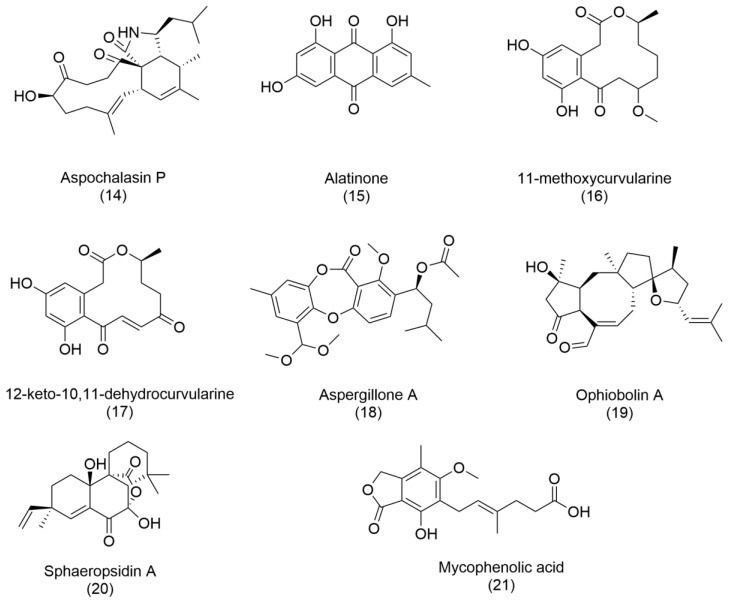
Structures of: aspochalasin P (**14**), alatinone (**15**), 11-methoxycurvularine (**16**), and 12-keto-10,11-dehydrocurvularine (**17**), aspergillone A (**18**), ophiobolin A (**19**) and sphaeropsidin A (**20**), mycophenolic acid (**21**).

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
