# Peer review of "The Significance of Fungal Specialized Metabolites in One Health Perspectives"

_ijms, 2025, doi:10.3390/ijms26073120_

Round 1

Reviewer 1 Report

Comments and Suggestions for Authors

The article covers key aspects such as host-fungal interaction, fungal metabolites, metabolite changes in the host, and the application of fungal metabolites as biomarkers and in personalized medicine.

The article is written in a very superficial manner, lacking solid data, recent references, a compelling argument, the author's perspective, and an outlook for the future.

Major scientific comments

  1. The article lacks a clear and streamlined structure.
  2. The article lacks a summarized table of major fungal metabolites demonstrating their potential as biomarkers in diagnosing.
  3. There is a need for an updated table of FDA-approved fungal metabolite drugs that are either available or currently undergoing clinical trials.
  4. The abstract and conclusion are contradictory, making it difficult to understand the article’s main message. It is unclear whether fungi are being presented as beneficial to human health due to their role as a source of various drugs or as harmful. Clarifying this inconsistency will help ensure a more coherent and impactful argument.
  5. Section 2 is missing citation/s.
  6. The discussion on human-fungal pathogen interaction lacks robust data and strong arguments. Strengthening this section with well-supported evidence and in-depth analysis.
  7. The overall review article lacks proper citations, despite the availability of hundreds of relevant papers on this topic. The absence of references significantly reduces the paper’s credibility, impact, and potential contribution to the field.

Author Response

The article covers key aspects such as host-fungal interaction, fungal metabolites, metabolite changes in the host, and the application of fungal metabolites as biomarkers and in personalized medicine.

The article is written in a very superficial manner, lacking solid data, recent references, a compelling argument, the author's perspective, and an outlook for the future.

A: We thank the reviewer for the time he spent in revising our opinion manuscript. However, our manuscript is not intended to be a comprehensive review but rather an opinion article, as it could also be read on the article-type, which presents the authors' perspectives based on their expertise in the field. It explores the emerging problem of fungal pathogens from a One Health perspective, starting from agriculture—a viewpoint not usually addressed in existing literature. We aim to highlight the interconnectedness of human, animal, and environmental health in the context of fungal diseases, emphasizing how lessons from agriculture can inform medical strategies. While we acknowledge that some aspects could be explored in greater detail, our focus is on stimulating discussion and proposing new ways of thinking about the topic. Nevertheless, we appreciate the feedback, and we answered the specific comments below; adding more recent references and more details were retained necessary by the authors in order to not go out from the style of an opinion manuscript.

Major scientific comments

  1. The article lacks a clear and streamlined structure.

A: Our manuscript is structured intentionally to align with the purpose of an opinion article. The three main sections—1. Learning from Agriculture, 2. Diagnostics of Fungal Metabolites, and 3. Therapeutic Applications of Fungal Metabolites—provide a logical progression that supports our perspective. This structure allows us to establish the relevance of agricultural insights, explore diagnostic challenges, and discuss therapeutic potential. We appreciate the feedback but believe the current structure aligns well with our intended focus.

2. The article lacks a summarized table of major fungal metabolites demonstrating their potential as biomarkers in diagnosing. There is a need for an updated table of FDA-approved fungal metabolite drugs that are either available or currently undergoing clinical trials.

A: We appreciate the suggestion; however, including a table summarizing major specialized fungal metabolites or FDA-approved fungal metabolite drugs would not align with the style and purpose of our opinion article. Instead of a table, we have incorporated figures highlighting relevant fungal metabolites, effectively illustrating key points while maintaining the article's narrative flow. While a detailed table would be valuable in a review article, our current approach is more suitable for an opinion article.

3. The abstract and conclusion are contradictory, making it difficult to understand the article’s main message. It is unclear whether fungi are being presented as beneficial to human health due to their role as a source of various drugs or as harmful. Clarifying this inconsistency will help ensure a more coherent and impactful argument.

A: Thank you for this comment. There is no contradiction between the abstract and conclusion; the key is understanding the dual nature of specialized fungal metabolites. These metabolites can harm infections and serve as therapeutic tools for cancer treatment. They can act as biomarkers for the early detection of fungal infections, leading to quicker diagnoses and more effective treatments. As is often the science case, the perspective from which we view these metabolites—whether as tools for medicine or as virulence factors in infections—shapes how we interpret their role in human health.

4. Section 2 is missing citation/s.

A: We incorporated proper citations in section 2, "Learning from Agriculture," and further developed the argumentation.

5. The discussion on human-fungal pathogen interaction lacks robust data and strong arguments. Strengthening this section with well-supported evidence and in-depth analysis.

A: We have further expaneded the human-fungal pathogen interaction discussion in line 108-123 and a recent review on the topic was added to the reference list

6. The overall review article lacks proper citations, despite the availability of hundreds of relevant papers on this topic. The absence of references significantly reduces the paper’s credibility, impact, and potential contribution to the field.

A. As outlined in our general response, our manuscript is not intended to be a comprehensive review; rather, it is an opinion article reflecting the authors' perspectives based on their expertise in the field. Notably, there are currently no relevant articles that explore fungal specialized metabolites from a One Health perspective. Traditionally, specialized metabolites have been studied in isolation within specific disciplines. Our goal is to emphasize the broader implications and potential synergies of studying fungal specialized metabolites, advocating for a more integrated and holistic approach to this topic, especially in light of contemporary challenges affecting human, animal, and environmental health.

Reviewer 2 Report

Comments and Suggestions for Authors

Dear Authors,

This manuscript described the fungal specialized metabolites for the significances in one-health perspectives. The manuscript was well written. Two points that need to be modified are mentioned below.

1) I suggest that the compounds in the text and Figures 1 and 2 be numbered. So the readers can easily find the structures of the compounds.

2) p.381, References, the magazine name should be unified. For example, the word JoF is the abbreviation of Journal of Fungi. It should be written as J. Fungi.

Author Response

Dear Authors,

This manuscript described the fungal specialized metabolites for the significances in one-health perspectives. The manuscript was well written. Two points that need to be modified are mentioned below.

A: We thank the reviewer for the positive evaluation of our opinion manuscript

1. I suggest that the compounds in the text and Figures 1 and 2 be numbered. So the readers can easily find the structures of the compounds.

A: We have added the numbers to the compounds in Figures 1 and 2 and along the text.

2) p.381, References, the magazine name should be unified. For example, the word JoF is the abbreviation of Journal of Fungi. It should be written as J. Fungi.

A: The reference style has been unified.

Reviewer 3 Report

Comments and Suggestions for Authors

In section 2. Learning from Agriculture, It is important to expand the explanation of mycotoxins especially aflatoxins, ochratoxins, and fumonisins (with representative structures) as the emerged from storage fungi on agricultural crops and have a bad health impact on both animals and human.  

In section 3. Diagnostics of Fungal Metabolites, It is important to refer to mucormycosis as emerged serious infection related to SARS-Cov-2.

In section 4. Therapeutic Applications of Fungal Metabolites, It is important to refer to marine environment as a habitat for diverse fungal species with potentials for producing novel bioactive fungal metabolites.

Line 52: “(http://www.WHO-OHHLEP)”. Please, provide the accession date. State the website in the references list.

Line 118: “Candida” make it italic.

Lines 123-125: “Aspergillus” make it italic.

Line 135: Revise “Candida albicans, Candida glabrata, Candida krusei, and Candida

tropicalis.” to “Candida albicans, C. glabrata, C. krusei, and C. tropicalis.”

Lines 147-150: “ex vivo”, “in vivo”, “in vitro”, “Cryptococcus neoformans”   make them italic.

Line 149: “Ephitelial Cells” to “ephitelial cells”.

Line 152: Revise “Cryptococcus neoformans”  to “C. neoformans

Line 248:  Revise “Figure 1 and Figure 2” to “Figures 1 and 2”. As they were stated once, please delete their mention from the upcoming paragraphs.

Line 297: Revise “(Bentley 2000)” to “[32]” as stated in references list.

Author Response

In section 2. Learning from Agriculture, It is important to expand the explanation of mycotoxins especially aflatoxins, ochratoxins, and fumonisins (with representative structures) as the emerged from storage fungi on agricultural crops and have a bad health impact on both animals and human.  

A: The explanation of mycotoxins has been expanded in the section, and a figure, new Figure 1, containing the general structure for the class, has been added.

In section 3. Diagnostics of Fungal Metabolites, It is important to refer to mucormycosis as emerged serious infection related to SARS-Cov-2.

A: This aspect is now discussed in the text of the section and a proper reference is now cited in the reference list.

In section 4. Therapeutic Applications of Fungal Metabolites, It is important to refer to marine environment as a habitat for diverse fungal species with potentials for producing novel bioactive fungal metabolites.

A: This aspect is now discussed in the text of the section and a proper reference is now cited in the reference list.

Line 52: “(http://www.WHO-OHHLEP)”. Please, provide the accession date. State the website in the references list.

A: The website and the accession date have been added to the reference list.

Line 118: “Candida” make it italic.

A:Corrected as suggested

Lines 123-125: “Aspergillus” make it italic.

A:Corrected as suggested

Line 135: Revise “Candida albicansCandida glabrataCandida krusei, and Candida

tropicalis.” to “Candida albicansC. glabrataC. krusei, and C. tropicalis.”

A: Corrected as suggested

Lines 147-150: “ex vivo”, “in vivo”, “in vitro”, “Cryptococcus neoformans”   make them italic.

A: Corrected as suggested

Line 149: “Ephitelial Cells” to “ephitelial cells”.

A: Corrected as suggested

Line 152: Revise “Cryptococcus neoformans”  to “C. neoformans

A: Corrected as suggested

Line 248:  Revise “Figure 1 and Figure 2” to “Figures 1 and 2”. As they were stated once, please delete their mention from the upcoming paragraphs.

A: Corrected as suggested

Line 297: Revise “(Bentley 2000)” to “[32]” as stated in references list.

A. Corrected as suggested

Round 2

Reviewer 1 Report

Comments and Suggestions for Authors

The author has revised the article based on the suggestions. However, a few were not incorporated, as this is an opinion article rather than a comprehensive article and I agree with the author. The article is suitable for publication in its current form.

Reviewer 3 Report

Comments and Suggestions for Authors

Thanks for providing the appropriate changes.